# Wide transition-state ensemble as key component for enzyme catalysis

**Gabriel E Jara**[1†]**, Francesco Pontiggia**[2‡]**, Renee Otten**[2§]**, Roman V Agafonov**[2#]**, Marcelo A Martí**[3*]**, Dorothee Kern**[2*¶]

[1]Departamento de Química Inorgánica, Analítica y Química-Física (INQUIMAE-CONICET), Universidad de Buenos Aires, Buenos Aires, Argentina; [2]Howard Hughes Medical Institute, Department of Biochemistry, Brandeis University, Waltham, United States; [3]Departamento de Química Biológica (IQUIBICEN-CONICET), Facultad de Ciencias Exactas y Naturales, Universidad de Buenos Aires, Buenos Aires, Argentina

**\*For correspondence:**
marti.marcelo@gmail.com
(MAM);
dkern@brandeis.edu (DK)

**Present address:** [†]Brazilian
Biosciences National Laboratory
(LNBio), Brazilian Center
for Research in Energy and
Materials (CNPEM), Campinas,
Brazil; [‡]Psivant Therapeutics,
Salem, United States; [§]Treeline
Biosciences, Watertown, United
States; [#]C4 Therapeutics,
Watertown, United States;
[¶]Department of Integrative
Structural & Computational
Biology and HHMI, The Scripps
Research Institute, La Jolla,
United States

**Competing interest:** See page
16

**Reviewing Editor:** Qiang Cui,
Boston University, United States

## eLife Assessment

In this potentially **important** study, the authors report results of QM/MM simulations and kinetic measurements for the phosphoryl-transfer step in adenylate kinase. The results point to the mechanistic proposal that the transition state ensemble is broader in the most efficient form of the enzyme (i.e., in the presence of Mg2+ in the active site) and thus a different activation entropy. With a broad set of computations and experimental analyses, the level of evidence is considered **solid** by some reviewers. On the other hand, there remain limitations in the computational analyses, especially regarding free energy profiles using different methodologies (shape of free energy profiles with DFTB vs. PBE QM/MM, and barriers with steered MD and umbrella sampling) and the activation entropy, leading some reviewers to the evaluation that the level of evidence is **incomplete**.

**Abstract** Transition-state (TS) theory has provided the theoretical framework to explain the enormous rate accelerations of chemical reactions by enzymes. Given that proteins display large ensembles of conformations, unique TSs would pose a huge entropic bottleneck for enzyme catalysis. To shed light on this question, we studied the nature of the enzymatic TS for the phosphoryl-transfer step in adenylate kinase by quantum-mechanics/molecular-mechanics calculations. We find a structurally wide set of energetically equivalent configurations that lie along the reaction coordinate and hence a broad transition-state ensemble (TSE). A conformationally delocalized ensemble, including asymmetric TSs, is rooted in the macroscopic nature of the enzyme. The computational results are buttressed by enzyme kinetics experiments that confirm the decrease of the entropy of activation predicted from such wide TSE. TSEs as a key for efficient enzyme catalysis further boosts a unifying concept for protein folding and conformational transitions underlying protein function.

## Introduction

Understanding the underlying physical mechanism of the impressive rate accelerations achieved by enzymes has been a central question in biology. Transition-state (TS) theory has provided the fundamental framework since the rate of a reaction is dictated by the free-energy difference between the ground state and TS (*Truhlar, 2015*). A wealth of research has led to the general notion that enzymatic rate acceleration is due to the enzymes' much higher affinity to the TS relative to its substrates. This idea has been supported experimentally by the high affinities measured for transition-state analogs (TSA) relative to substrates, leading to the design of TSA as high-affine enzyme inhibitors (*Lienhard, 1973*; *Schramm, 2007*). Since the TS represents a maximum in the free-energy landscape, its structure

cannot be determined directly due to its low probability and ephemeral nature, TSA-bound species have been extensively used as structural proxies.

The classic TS theory and structural visualization of enzyme/TSA complexes lead to a current view of quite unique structures on the dividing surface at the maximum in the free-energy landscape. In contrast, it is now generally accepted that proteins are large ensembles of conformations, a concept rooted in pioneering work by Frauenfelder and coworkers for protein function, in analogy to the ensemble/funnel concept of protein folding (*Frauenfelder and Wolynes, 1985*). Consequently, enzyme–substrate (ES) complexes – experimentally accessible as minima in the free-energy landscape – are composed of a vast ensemble of molecular configurations due to their macromolecular nature. Hence, catalysis by pathing through exclusive TS's would pose a huge entropic bottleneck/energy barrier for enzyme-catalyzed reactions.

Motivated by this consideration, together with the marginal outcomes in current enzyme design relative to the catalytic power of naturally evolved enzymes, we set out to investigate the atomistic nature of TSs of the chemical step in an enzymatic cycle and its implication for rate acceleration by combining quantum-mechanics/molecular-mechanics (QM/MM) calculations and experiments. We take advantage of the groundbreaking work on TS theory by QM simulation developed in the field (*Kamerlin et al., 2013*; *Masgrau and Truhlar, 2015*; *Warshel and Bora, 2016*; *Zinovjev and Tuñón, 2017*), well summarized in several reviews (*Cui, 2016*; *Senn and Thiel, 2009*; *van der Kamp and Mulholland, 2013*). We chose a phosphoryl-transfer (P-transfer) reaction due to its impressive enzymatic rate acceleration, as the uncatalyzed reactions have extremely high-energy barriers (*Kerns et al., 2015*; *Lassila et al., 2011*). The chemical rationale for these high barriers and reasons as to why P-transfer reactions are ubiquitous in living organisms and are involved in almost all essential biological processes is elegantly discussed in a fundamental paper by Westheimer in 1987: 'Why nature chose phosphates' (*Westheimer, 1987*). The central role of enzyme-catalyzed P-transfer reactions in biology (i.e., genetic code, energy storage, signaling, and compartmentalization) has led to extensive discussions and controversies of how enzymes catalyze these vital reactions so efficiently (*Allen and Dunaway-Mariano, 2016*; *Kamerlin et al., 2013*; *Kerns et al., 2015*; *Lassila et al., 2011*; *Pabis et al., 2016*; *Westheimer, 1987*). Here, we combine QM/MM calculations with experimental temperature- and pH-dependent kinetic studies and X-ray crystallography of adenylate kinase (Adk) to shed light on the question of TS structures for the chemical step in its catalytic cycle. While QM/MM methods have been extensively applied to study enzymatic reactions, yielding accurate and detailed descriptions of the molecular events during catalysis (*Acevedo and Jorgensen, 2010*; *Cheng et al., 2005*; *Dinner et al., 2001*; *Hahn et al., 2015*; *Hirvonen et al., 2020*; *Karplus and Kuriyan, 2005*; *Lai and Cui, 2020a*; *López-Canut et al., 2011*; *Mones et al., 2013*; *Palermo et al., 2015*; *Rosta et al., 2014*; *Roston et al., 2018*; *Roston and Cui, 2016*; *Schwartz and Schramm, 2009*; *Senn and Thiel, 2009*; *Turjanski et al., 2009*), we test here the reaction free-energy profiles (FEPs) from simulations subsequently by designed experiments. We deliver a revised notion of TS stabilization by the enzyme through our discovery of a structurally broad transition-state ensemble (TSE).

## Results

Adk is an extensively studied phosphotransferase that catalyzes the reversible, roughly isoenergetic conversion of two ADP molecules into ATP and AMP (*Figure 1A, B*), thereby maintaining the cellular concentrations of these nucleotides (*Dzeja and Terzic, 2009*). It is an essential enzyme found in every cell and organism. Extensive structural, dynamic, and kinetic studies led to a comprehensive picture of the overall Adk reaction mechanism and its underlying energy landscape (*Figure 1A*; *Beckstein et al., 2009*; *Berry et al., 1994*; *Berry et al., 2006*; *Henzler-Wildman et al., 2007*; *Kerns et al., 2015*; *Müller et al., 1996*; *Müller and Schulz, 1992*; *Wolf-Watz et al., 2004*). The conformational change of the AMP- and ATP-lids closing and opening is crucial in the enzymatic cycle: Lid-closing positions the two substrates and the active-site residues for efficient chemistry (i.e., P-transfer) and prohibits the alternative, energetically favorable reaction of phosphoryl hydrolysis (*Kerns et al., 2015*). Lid-opening, essential for product release, and not P-transfer is the rate-limiting step in the catalytic cycle and $Mg^{2+}$ accelerates both steps (*Kerns et al., 2015*). Here, we investigate the actual P-transfer step computationally, followed by experimental testing of our computational results. This chemical step would take about 7000 years without the enzyme (*Stockbridge and Wolfenden, 2009*) compared to >5000 $s^{-1}$ with the enzyme (*Kerns et al., 2015*).

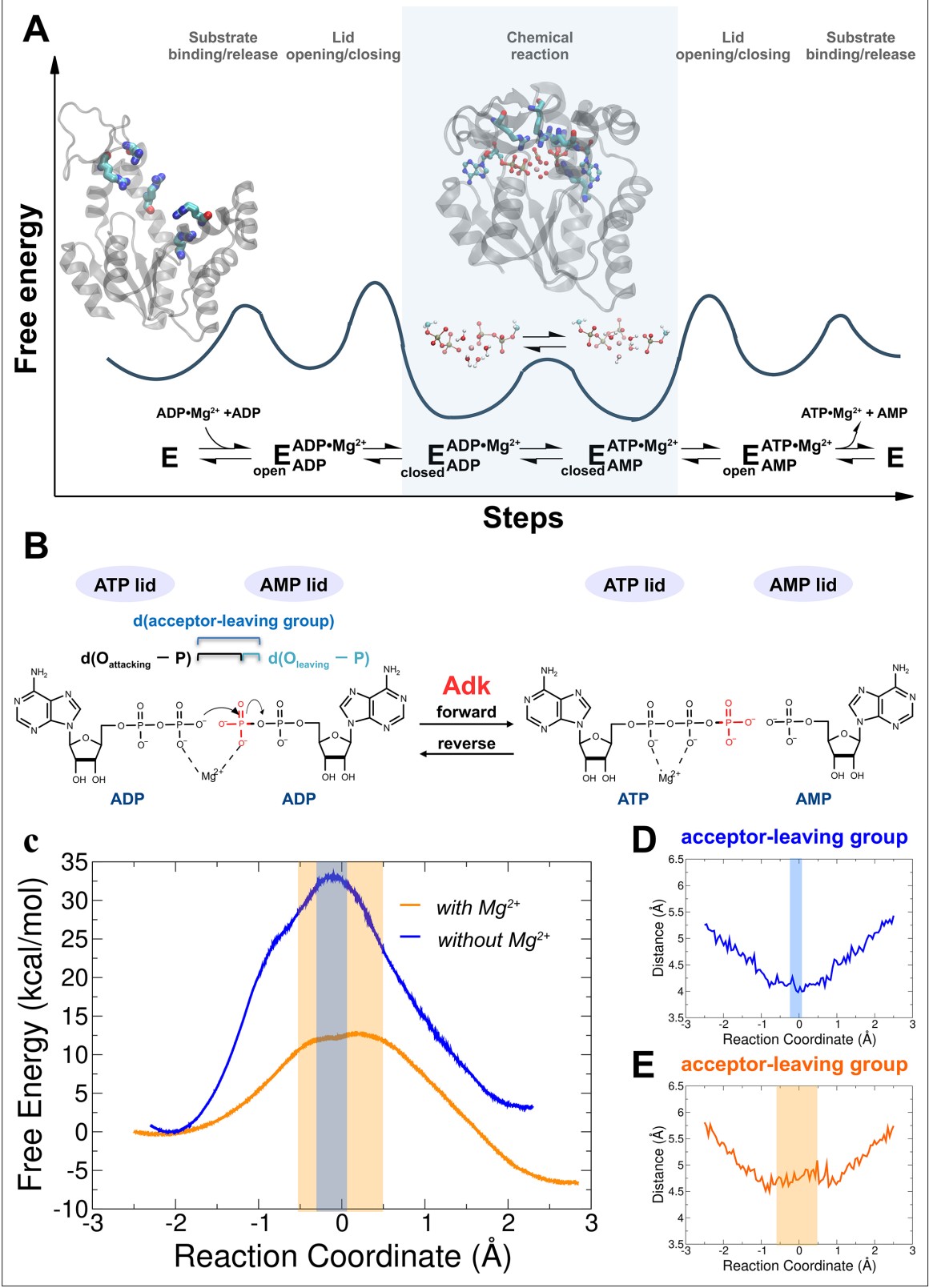

**Figure 1.** Investigation of the chemical step of phosphoryl transfer by quantum-mechanics/molecular-mechanics (QM/MM) calculations in the enzymatic reaction of adenylate kinase (Adk). (**A**) Complete reaction scheme with corresponding illustrative free-energy landscape highlighting the chemical phosphoryl-transfer step (modified from **Kerns et al., 2015**). Protein structures shown are apo Adk in the open conformation and Adk in the closed conformation with two bound ADP molecules and one $Mg^{2+}$ atom (active-site arginine side chains are shown in stick representation). (**B**) Phosphoryl-

*Figure 1 continued on next page*

*Figure 1 continued*

transfer step is drawn with corresponding distances used to define the reaction coordinate as used in panels D, E. (**C**) Free-energy profiles for the Adk-catalyzed interconversion of ADP–ADP into ATP–AMP in the absence (blue) and presence of $Mg^{2+}$ (orange) from QM/MM calculations. AD(T)P is fully charged for the reaction with $Mg^{2+}$, and singly protonated on one ADP β-oxygen for reaction without $Mg^{2+}$. The reaction coordinate is defined as the difference between the distance of the leaving oxygen to the transferring phosphorus $d(O_{leaving} - P)$ and the distance of the attacking oxygen to the transferring phosphorus $d(O_{attacking} - P)$. Distance between acceptor and leaving oxygens along the reaction coordinate in the presence (**E**) and in the absence (**D**) of $Mg^{2+}$. The transition-state regions in C–E are highlighted in orange and gray, respectively.

The online version of this article includes the following figure supplement(s) for figure 1:

**Figure supplement 1.** Quantum-mechanics/molecular-mechanics (QM/MM) system.

**Figure supplement 2.** Superposition of the starting model structure for quantum-mechanics/molecular-mechanics (QM/MM) and the crystallographic structure (PDB ID: 4CF7; *Kerns et al., 2015*) of adenylate kinase (Adk) with two ADP/Mg bound.

**Figure supplement 3.** Forward and backward quantum-mechanics/molecular-mechanics (QM/MM) reactions for adenylate kinase (Adk) with two ADP and $Mg^{2+}$ bound.

**Figure supplement 4.** Forward and backward quantum-mechanics/molecular-mechanics (QM/MM) reactions for adenylate kinase (Adk) with ADP, ADP, and $Mg^{2+}$ bound.

**Figure supplement 5.** Forward and backward quantum-mechanics/molecular-mechanics (QM/MM) reactions for adenylate kinase (Adk) with two ADP and without $Mg^{2+}$ bound.

**Figure supplement 6.** Free-energy profiles for the adenylate kinase (Adk)-catalyzed interconversion of ADP–ADP into ATP–AMP and presence of $Mg^{2+}$ employing higher-level DFT(PBE) free-energy calculations (see methods for details).

## QM/MM calculations of the P-transfer step in Adk

In order to shed light on the mechanism by which Adk from *Aquifex aeolicus* catalyzes the phosphoryl-transfer reaction by more than 12 orders of magnitude (*Kerns et al., 2015*), we performed QM/MM simulations starting with two ADP molecules and $Mg^{2+}$ bound in the active site. The starting structures for the simulations were prepared using the X-ray structure of Adk in complex with Ap5A (P1,P5-Di(adenosine-5′) pentaphosphate) and coordinated to $Zn^{2+}$ (2RGX; *Henzler-Wildman et al., 2007*). ADP–ADP coordinates were built using Ap5A as a template and $Zn^{2+}$ was replaced by $Mg^{2+}$. For the QM/MM simulations, the QM region was defined as the diphosphate moiety of both ADP molecules, the $Mg^{2+}$ ion, plus the four coordinating water molecules. The rest of the system was described at molecular-mechanics level using the AMBER ff99sb force field (*Hornak et al., 2006*) and was solvated with TIP3P water molecules (*Figure 1—figure supplement 1*).

The equilibrated starting structures agree well with the X-ray structures of the enzyme bound to $Mg^{2+}$/ADP (4CF7; *Kerns et al., 2015*; *Figure 1—figure supplement 2*). Steered molecular dynamics simulations were run in both the forward (ADP/ADP to ATP/AMP) and reverse direction (ATP/AMP to ADP/ADP) with $Mg^{2+}$ present in the active site (*Figure 1—figure supplement 3*). Since it is unknown whether the fully charged or monoprotonated nucleotide (on one β-ADP oxygen) state is the more reactive configuration, we performed QM/MM simulations for both cases. FEPs of the P-transfer were

**Table 1.** Free-energy profile estimates of the free-energy parameters of the reaction, using self-consistent charge-density functional tight-binding (SCC-DFTB) (all values in kcal/mol).

| | With $Mg^{2+}$ | Without $Mg^{2+}$ | With $Mg^{2+}$, ADP monoprotonated |
|---|---|---|---|
| $\Delta G^{*}$ | −6 (1.7) | +4 (2.5) | +6 (1.9) |
| $\Delta_{r}G^{\dagger}$ | 13 (0.9) | 34 (1.6) | 23 (0.9) |
| $\Delta_{b}G^{\dagger}$ | 20 (0.8) | 30 (0.9) | 18 (0.9) |
| $\xi$(TS) | −0.5 to 0.7 [‡]<br>0.0 to 0.2 | ▶ −0.2 to 0 [‡]<br>▶ −0.1 | −0.3 to −0.1 [‡]<br>Not calculated |
| $\zeta$(TS) | 165 to 190 | 165 to 180 | 170 to 180 |

$\xi$(TS) is the range of the reaction coordinate in the TSE (in Å); $\zeta$(TS) is the improper dihedral angle of the transferring phosphate in the TS. The estimated errors of the free energies are in parenthesis and are computed as described in Materials and methods.

[*]overall reaction free energy.

[†]activation free energy of the forward reaction; activation free energy of the backward reaction.

[‡]Values were analyzed by a visual analysis of the multiple steered molecular dynamics (MSMD) trajectories, no committor distribution were calculated.

determined using Multiple Steered Molecular Dynamics and Jarzynski's Relationship (*Crespo et al., 2005*; *Jarzynski, 1997*; *Ramírez et al., 2014*; *Figure 1C*, *Figure 1—figure supplements 3–5*). The results reveal a much smaller free energy of activation for the fully charged nucleotide state ($\Delta_f G^{\ddagger}$ of 13 ± 0.9 kcal/mol) relative to monoprotonated state ($\Delta_f G^{\ddagger}$ of 23 ± 0.9 kcal/mol) (*Table 1*); hence, we conclude that this is the most reactive enzyme configuration.

When the calculations were repeated in the absence of $Mg^{2+}$ (with the nucleotide monoprotonated, since the fully charged nucleotide state prohibited the reaction), a large increase in the free-energy activation barrier was observed relative to the $Mg^{2+}$-bound system (*Figure 1C*, *Figure 1—figure supplement 5*, $\Delta_f G^{\ddagger}$ of 34 ± 1.6 kcal/mol), in agreement with expectations from experiments (*Kerns et al., 2015*). We note that only a lower limit for the overall acceleration by $Mg^{2+}$ (>$10^5$-fold) could be estimated from published results (*Kerns et al., 2015*), since the P-transfer was too fast to be measured experimentally in the presence of $Mg^{2+}$.

During revision, we tested the accuracy and robustness of our results from our original QM/MM level of theory by employing higher-level pure DFT(PBE) free-energy calculations similar to the approaches used by *Ganguly et al., 2020*; *Figure 1—figure supplement 6*. The results are in excellent agreement with the FEPs obtained before (*Figure 1C, E*; *Figure 2A, C*) thus further supporting the computational results.

## The TSE – transferring phosphoryl group delocalized

The nature of the TS of enzyme-catalyzed P-transfer reaction with respect to its associative and dissociative character has been of significant interest and heated debate (*Kamerlin et al., 2013*; *Kamerlin and Wilkie, 2007*; *Lassila et al., 2011*; *Roston and Cui, 2016*). The definition as well as theoretical and experimental approaches to distinguish between them are rooted in elegant and fundamental work on *nonenzymatic* P-transfer reactions (*Duarte et al., 2015*; *Hengge, 2002*; *Hou et al., 2012*; *Hou and Cui, 2012*; *Kamerlin et al., 2013*; *Kerns et al., 2015*; *Kirby and Nome, 2015*; *Stockbridge and Wolfenden, 2009*). For Adk, a concerted mechanism is observed with no intermediates (*Figures 1 and 2*). Instead of using the classification of associative versus dissociative character (*Lassila et al., 2011*), we rather apply the clear definition by Cui and coworkers of a tight versus loose TS for concerted P-transfer reactions as they are unambiguously related to the bond-order in the TS (*Lai and Cui, 2020a*; *Lai and Cui, 2020b*; *Roston et al., 2018*; *Roston and Cui, 2016*). For Adk, a contraction of the active site is observed while passing through the TSE, as captured by the decrease in the distance between acceptor and leaving group during the transition (*Figure 1D, E*). In the absence of $Mg^{2+}$, the acceptor and donor need to be within 4 Å for the reaction to occur (*Figure 1D*) whereas in the presence of $Mg^{2+}$ an acceptor–donor distance of 4.5 Å is sufficient (*Figure 1E*). The character of the TSE can best be seen from the widely used Moore-O'Ferrall–Jencks diagram (*Jencks, 1972*; *O'Ferrall, 1970*) that plots the distance from the transferring phosphate to the oxygen of the donor and acceptor, respectively (*Figure 2*). The TSE changes from tight/synchronous without $Mg^{2+}$ (meaning high bond-order for both bonds between transferring phosphate to the leaving group and the attacking nucleophile) into loose with $Mg^{2+}$ (lower bond-order for these two bonds; *Figures 2 and 3*; *Lai and Cui, 2020a*; *Lai and Cui, 2020b*; *Roston et al., 2018*; *Roston and Cui, 2016*).

Importantly, from these two-dimensional (*Figure 2*) and the one-dimensional (*Figure 1C*) free-energy plots, we noticed a novel striking feature: a large ensemble of conformations in the TS region with vast differences in the position of the transferring phosphate but equal values in free energy for the enzyme with $Mg^{2+}$ (light blue dots in *Figure 2*, yellow area in *Figure 1C*). In other words, the enzyme seems to operate with a wide TSE. In contrast, without $Mg^{2+}$ fewer conformations seem to comprise the TSE. To dive more into this unexpected computational result and better visualize the TSE, we show zoom-ins of representative TSE snapshots and their superposition (*Figure 3*, *Figure 3—figure supplement 1*). Notably, the TSE conformations are distributed along the reaction coordinate. The difference in position of the transferring phosphate along the reaction coordinate within the TSE (about 1 Å with $Mg^{2+}$; *Figure 1c*, *Figure 2* and *Figure 3b*) implies that the TSE contains many highly asymmetric conformations, meaning that the transferring phosphate can be much closer to the leaving oxygen than the attacking oxygen and vice versa (*Figure 3D*). This asymmetry is logically tied with nonplanar configurations of the transferring phosphate (*Figure 3D* and *Table 1*).

Comparison of the TSE snapshots for the fully active, $Mg^{2+}$-bound enzyme obtained from QM/MM with an X-ray structure of the same enzyme in complex with the TSA (ADP–$Mg^{2+}$–$AlF_4$–AMP) (*Kerns*

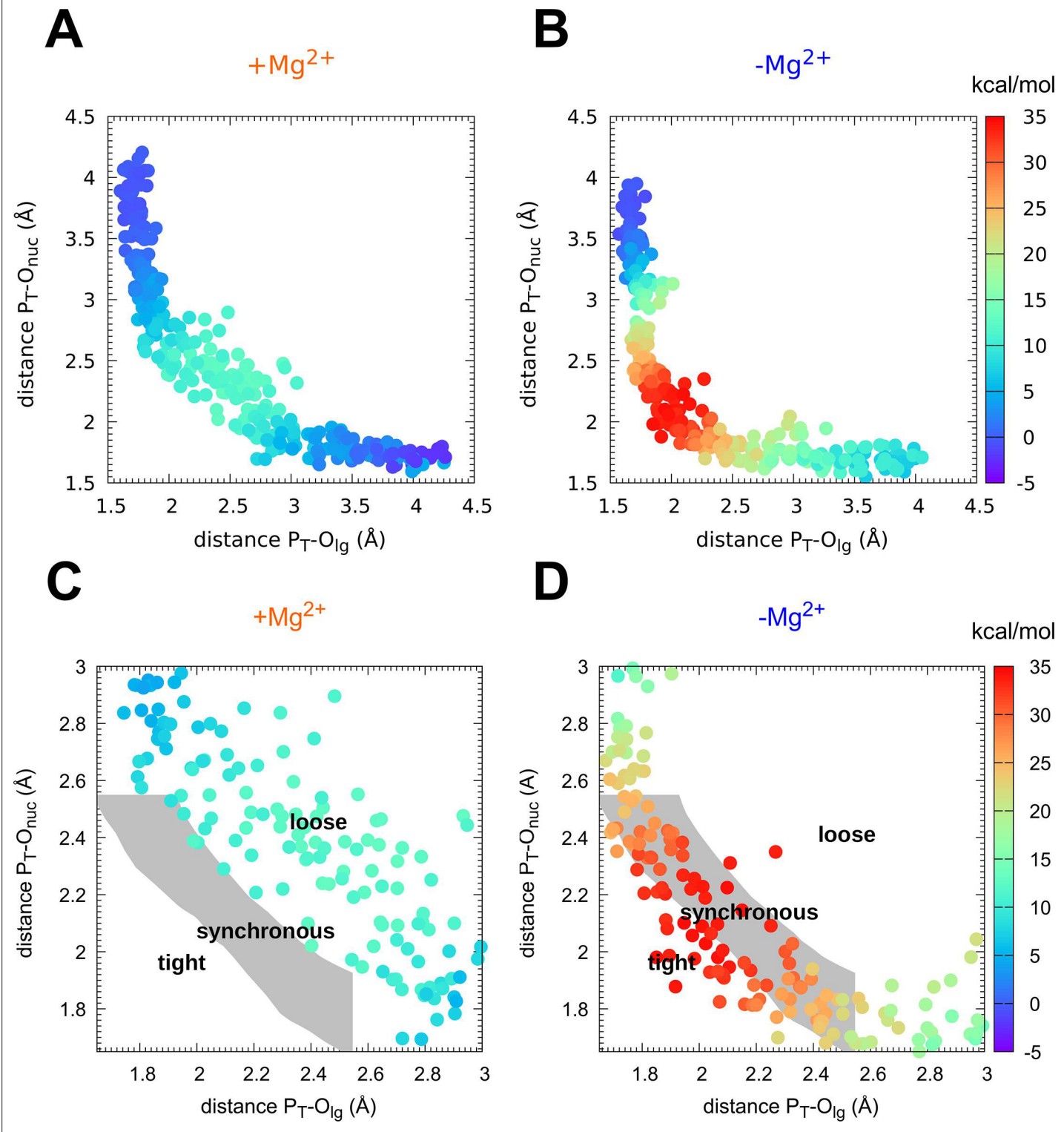

**Figure 2.** Mechanism of phosphoryl transfer. Diagram of (*Jencks, 1972*; *O'Ferrall, 1970*) from the quantum-mechanics/molecular-mechanics (QM/MM) simulations, plotting the two P–O distances involved in the P-transfer for the reaction (**A, C**) with $Mg^{2+}$ and (**B, D**) without $Mg^{2+}$ ($P_T$ for transferring phosphate). The theoretical transition pathways for a tight, synchronous, and loose transition state are shown in C, D (as defined in *Roston and Cui, 2016*).

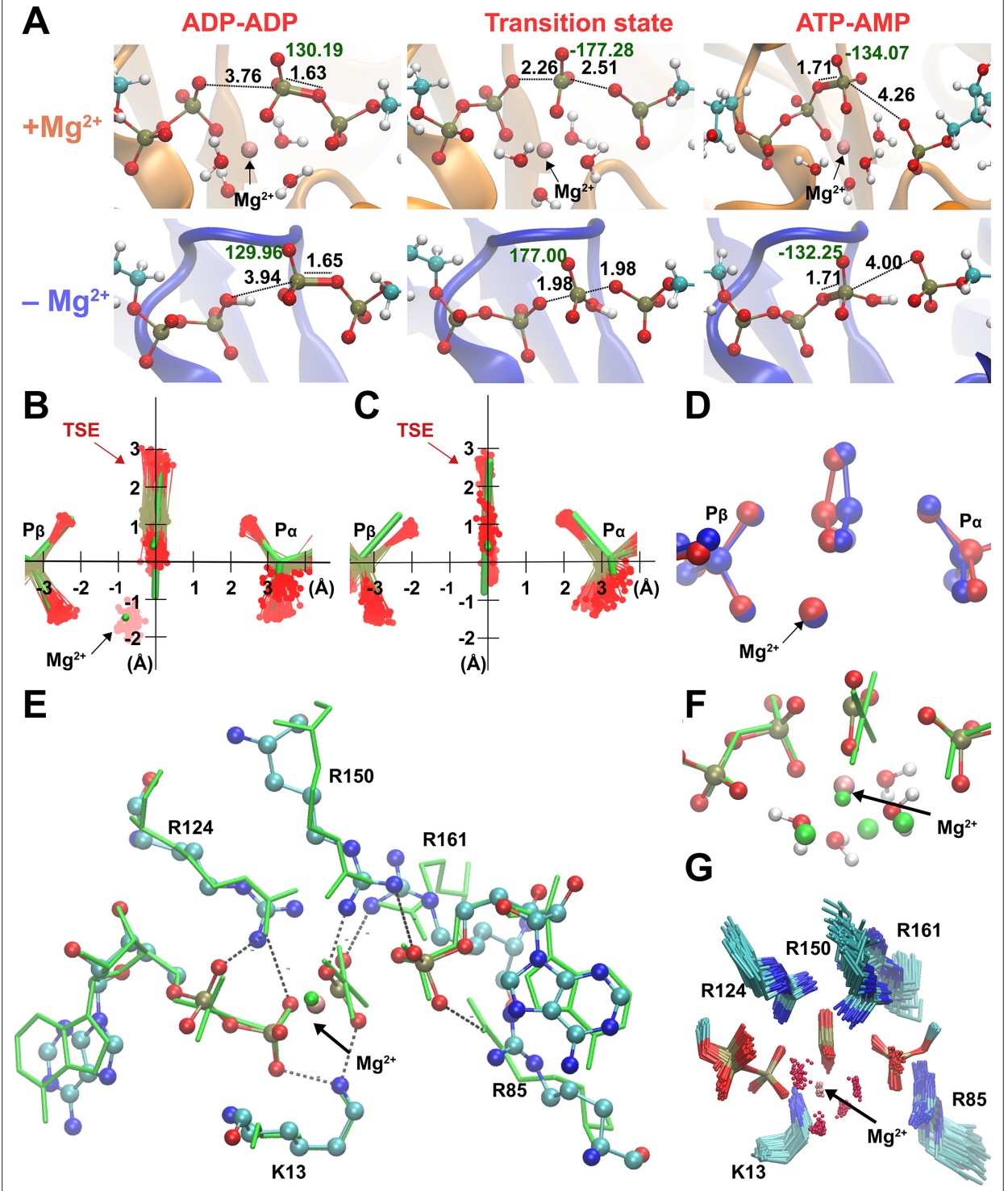

**Figure 3.** Broad transition-state ensemble (TSE) in fully active enzyme. (**A**) Representative snapshots for structure of reactants, transition states, and products in the adenylate kinase (Adk) active site in the presence and absence of magnesium. $d(O_{attacking} - P)$ and $d(O_{leaving} - P)$ are shown. Labels in black indicate the length of the bonds involved in the phosphate transfer and in green, the dihedral angle of the phosphoryl group. Superposition of the TSEs reveals a wider TSE with $Mg^{2+}$ (**B**) relative to the one without the cation (**C**) with mean rmsd (± SD) of distances of the central P atom from its 'average' position of 0.30 ± 0.11 Å (**B**) and 0.13 ± 0.06 Å (**C**). The TSEs are superimposed with the X-ray structure solved with a transition-state analog in green (AMP, $AlF_4^-$, and ADP) reported in *Kerns et al., 2015* (PDB ID: 3SR0). (**D**) Superposition of two extreme structures out of the large TSE for the enzyme with $Mg^{2+}$, one where the phosphoryl group is closest to the donor oxygen (blue) and the other closest to the acceptor oxygen (red) highlighting the asymmetric character of TSE members. (**E, F**) Superposition of most symmetric snapshot from TSE of quantum-mechanics/molecular-mechanics (QM/

*Figure 3 continued on next page*

*Figure 3 continued*

MM) calculations (ball and stick representation in cyan) with the X-ray structure of transition-state analog (green, PDB ID: 3SR0 *Kerns et al., 2015* including coordinating water molecules in F). (**G**) Zoom into the active site to display the broad TSE in the presence of Mg$^{2+}$ aided by flexible Arg and Lys side chains in the active site.

The online version of this article includes the following figure supplement(s) for figure 3:

**Figure supplement 1.** Representative snapshots for the reactant, transition state (TS), and product states in the *monoprotonated system with Mg$^{2+}$*.

**Figure supplement 2.** A representative transition-state structure is compared against the X-ray structure PDB ID: 3SR0 (*Kerns et al., 2015*).

**Figure supplement 3.** Reaction with Mg$^{2+}$ and fully charged nucleotides.

**Figure supplement 4.** Reaction without Mg$^{2+}$.

**Figure supplement 5.** Reaction with Mg$^{2+}$ and protonated in an oxygen of the beta-phosphate of ADP$_{ATP-lid}$.

**Figure supplement 6.** Representative snapshots for the reactant, transition state (TS), and product states in the system with Mg$^{2+}$.

**Figure supplement 7.** Representation of the interactions between the nucleotides with the active-site amino acids and Mg$^{2+}$ in the active site of adenylate kinase (Adk) for a representative snapshot from the transition-state ensemble (TSE).

**Figure supplement 8.** Superposition of reactant (blue), transition state (TS) (red), and product (green) states focusing on the transferring phosphate and the main amino acids assisting phosphoryl transfer, for the system with Mg$^{2+}$.

**Figure supplement 9.** Committor distribution (histogram) between reaction coordinate values (RC) from −0.2 to 0.3 for the reaction with (**A**) and without Mg$^{2+}$ (**B**).

*et al., 2015*) serves as an initial experimental validation of our simulations (*Figure 3C, F, Figure 3— figure supplement 2*). At the same time, this comparison highlights the power of the QM/MM simulations to investigate the catalytic mechanism, as the TSA-bound X-ray structure seems to imply quite a unique TS conformation, in sharp contrast to the broad TSE discovered in our simulations.

A detailed analysis of the pathways with and without Mg$^{2+}$ reveals well-known features from other P-transfer enzymes, such as coordinated changes of the Mulliken charge populations coupled with proton transfers (*Figure 3—figure supplements 3–5*). Mg$^{2+}$ achieves its catalytic effect on the chemical step playing several roles. It enables the donor and acceptor phosphates to come close enough to react, thus achieving a reactive conformation. In the absence of Mg$^{2+}$, a proton must bridge the terminal phosphates to prevent the strong electrostatic repulsion between the two nucleotides. In addition, Mg$^{2+}$ stabilizes the charge of the transferring phosphate and thereby lowering the enthalpic barrier. Most interestingly, the presence of a cation seems to result in a wide TSE. Structures of the TSE show that Mg$^{2+}$ is optimally coordinated to the transferring phosphates and water molecules (*Figure 3G*).

Besides the multifaceted role of Mg$^{2+}$, our simulations provide insights into the function of the fully conserved arginine residues in the active site for lowering the activation barrier for this chemical reaction (*Figure 3—figure supplements 6–8*). R85 appears to arrange the beta-phosphates of the ADPs in the proper position for the reaction to be started. R150 and R161 are involved in anchoring ATP and AMP residues in the backward reaction. These last two residues and R85 interact with the Pα of AMP, stabilizing its negative charge. R36 is near to the alpha-phosphate of AD(M)P along the entire catalytic reaction, stabilizing both the TS and products. Overall, interactions between these arginine side chains and several backbone amides and the Mg$^{2+}$ with the phosphates of the substrate make up a well-organized, asymmetric active site enabling efficient reversible P-transfer with a wide TSE (*Figure 3G*).

## Umbrella sampling and committor analysis buttress wide TSE for Adk with Mg$^{2+}$

To verify our computational finding of a wide TSE in the fully assembled enzyme with Mg$^{2+}$ in contrast to a narrower TSE without a divalent metal, we performed QM/MM umbrella sampling and a committor analysis as described in methods. The umbrella sampling FEPs (*Figure 4a–d*) corroborate the key difference in the range of the TSE. Noteworthy, the activation free energies are smaller for the umbrella sampling when compared to the multiple steered molecular dynamics (MSMD). The largest difference being that for the reaction in the absence of Mg (*Figure 1C*). Higher barriers when comparing the results using Jarzynski's to those obtained with umbrella sampling is not totally

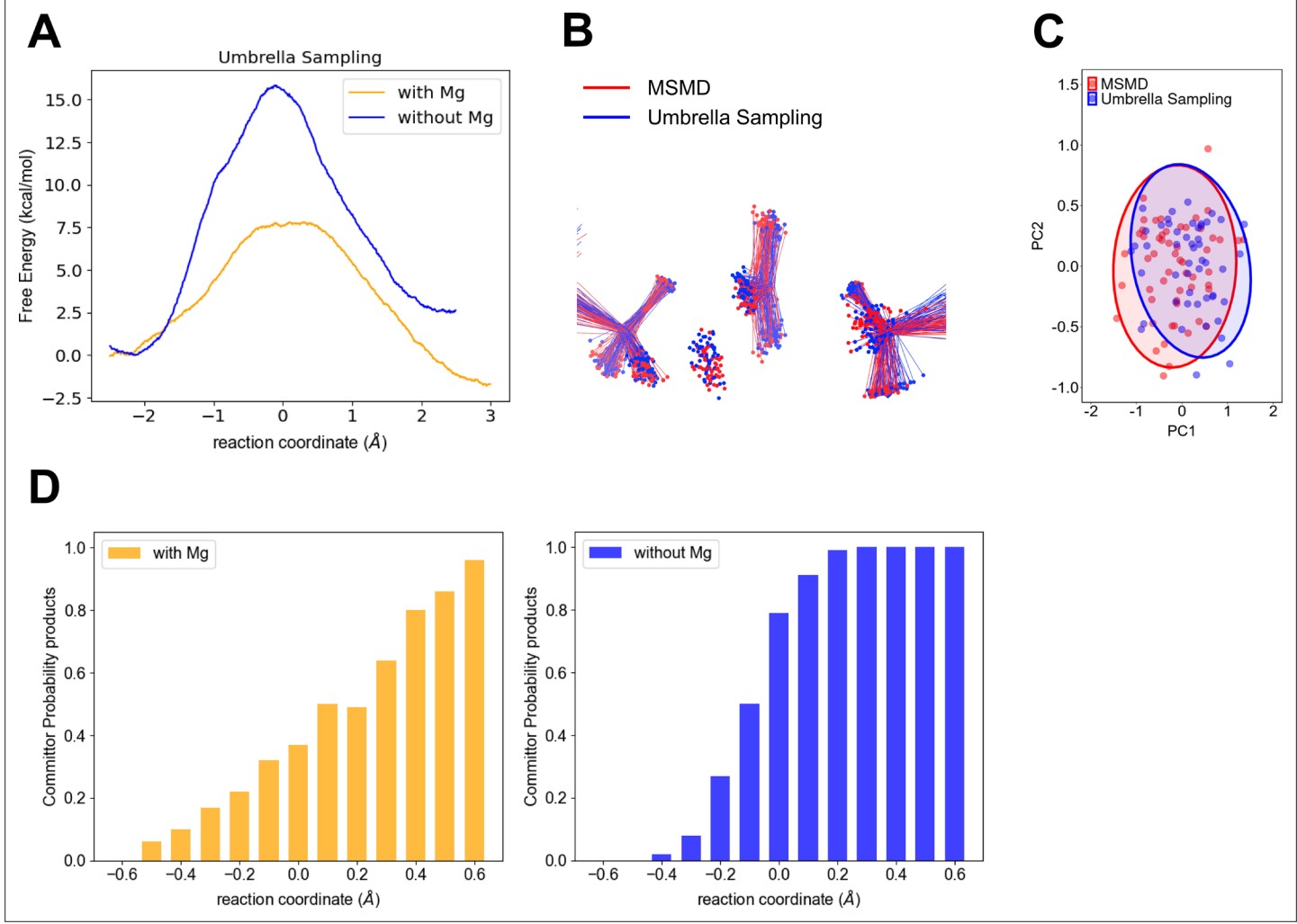

**Figure 4.** Computational testing of wider transition-state ensemble (TSE) for P-transfer step in adenylate kinase (Adk) in the presence of $Mg^{2+}$ versus no divalent metal. (A) Free-energy profiles obtained by umbrella sampling for reactions with and without $Mg^{2+}$. (B) Comparison of the TSE from multiple steered molecular dynamics (MSMD) and umbrella sampling for runs in the presence of $Mg^{2+}$. The TSE structures were aligned by atoms of the $P_\alpha$ and $O3\alpha$ atoms from AD(M)P and $P_\beta$ and $O3\beta$ atoms from AD(T)P. (C) First principal component (PC1) and second principal component (PC2) obtained from principal component analysis (PCA) for b. The PCA was calculated using the transferring phosphate atoms, $Mg^{2+}$ ion, $P\alpha$ and $O3\alpha$ atoms from AD(M)P and $P_\beta$ and $O3\beta$ atoms from AD(T)P. The TSE structures aligned by atoms of the $P-O_{leaving}$ and $P-O_{attacking}$ (see method for details). (D) Commitment plot with the committor probability to products (ATP + AMP) for the reaction with and without $Mg^{2+}$.

unexpected. It is well known that exponential averaging tends to overestimate free-energy barriers, and particularly when they are too high (*Park et al., 2003*).

Notably, the calculated difference in activation barriers with and without Mg from these umbrella simulations of about 9.5 kcal/mol is in good agreement with the experimentally determined ones of ≥11 kcal/mol (P-transfer with Mg is >500 $s^{-1}$ (*Kerns et al., 2015*) compared to $7.5 \times 10^{-4}$ $s^{-1}$ measured here).

A consequent comparative commitment analysis that calculates the probability of reaching either reactants or products displays a very shallow change along the reaction coordinate in the presence of $Mg^{2+}$, further corroborating a wide TSE for the P-transfer step in the fully active enzyme, in stark contrast to a narrow and steep change in the committors indicative of a narrower TSE without $Mg^{2+}$ (*Figure 4d*). The TSE is defined in function of the committor distribution (*Figure 3—figure supplement 9*), showing the TSE is at −0.1 Å of the reaction coordinate for the reaction without $Mg^{2+}$. In contrast, the TSE is wider for the reaction with $Mg^{2+}$ (reaction coordinate values in the range from 0.0 to 0.2 Å; *Figure 3—figure supplement 9* and *Table 1*).

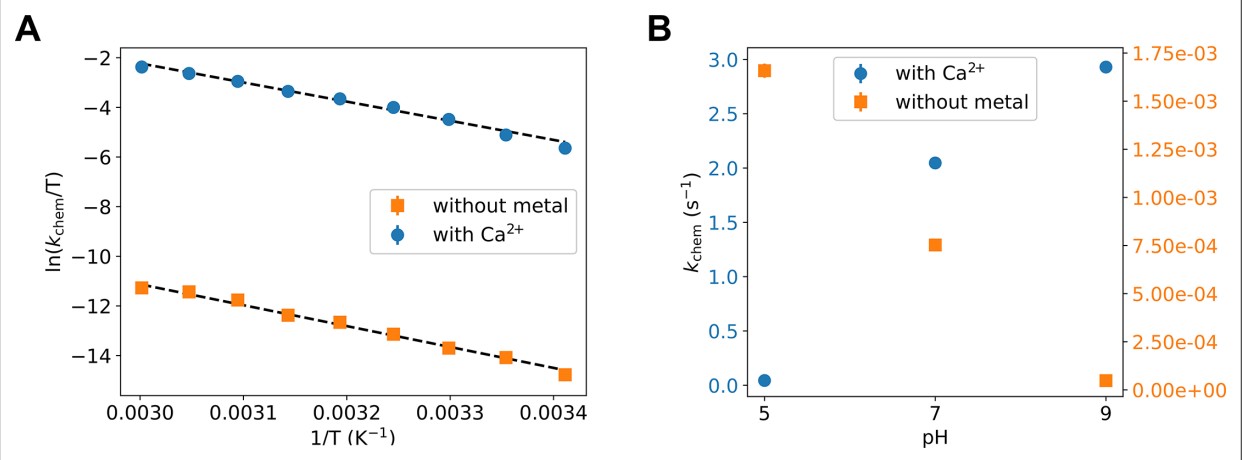

**Figure 5.** Experimental testing of the computational findings. (**A, B**) Turnover rate constants that represent the chemical step ($k_{chem}$) under these conditions were measured with 8 mM ADP/Mg$^{2+}$ by HPLC detection of build-up of ATP and AMP. (**A**) Temperature dependence of the phosphoryl-transfer step measured in presence of calcium and in absence of divalent any cations, plotted as Eyring plots. Fits to the Eyring equation (dashed lines) result in $\Delta H^{\ddagger}$ = 16.7 ± 1 kcal/mol and $\Delta S^{\ddagger}$ = −19.1 ± 3.1 cal/mol/K without metal, and $\Delta H^{\ddagger}$ = 15.3 ± 0.5 kcal/mol and $\Delta S^{\ddagger}$ = −5.7 ± 1.6 cal/mol/K with Ca$^{2+}$. (**B**) pH dependence of $k_{chem}$ measured in the presence of calcium and in absence of any divalent cation.

## Experimental characterization of the activation parameters of the P-transfer step

We felt the need to additionally experimentally test our major new finding from the QM/MM simulations: a delocalized TSE for the fully active enzyme. This feature would result in a lowering of the entropic barrier for the chemical step, thereby contributing to the enzyme-catalyzed rate enhancement. To avoid the issues of accuracy of QM/MM simulations that are well documented (*Acevedo and Jorgensen, 2010*; *Elstner, 2007*; *Gaus et al., 2014*; *Roston et al., 2018*), our system has the advantage of comparing the exact same reaction coordinate with the only difference being the presence/absence of a single atom, Mg$^{2+}$, thereby studying differences rather than absolute values. Since the reaction with Mg$^{2+}$ (fully active enzyme) revealed a more pronounced delocalization than the reaction in its absence, the divalent cation is predicted to lower the activation entropy. To test this prediction, we experimentally determined the enthalpic and entropic contributions to the chemical reaction barrier in Adk by measuring its temperature dependence in the presence and absence of Ca$^{2+}$. The experimental trick of replacing Mg$^{2+}$ with Ca$^{2+}$ was used since the chemical step with Mg$^{2+}$ is too fast to be experimentally measured, and previous studies showed that Ca$^{2+}$ is an appropriate mimic to selectively probe the chemical step (*Kerns et al., 2015*).

The resulting Eyring plots for the two experiments (*Figure 5A*) deliver the enthalpic ($\Delta H^{\ddagger}$ of 15.3 ± 0.5 and 16.7 ± 1 kcal/mol) and entropic ($\Delta S^{\ddagger}$ = −5.7 ± 1.6 and −19.1 ± 3.1 cal/mol/K) contributions to the chemical reaction barrier in the presence and absence of Ca$^{2+}$, respectively. The presence of Ca$^{2+}$ reduces both barriers considerably. The decrease in the enthalpic barrier by Ca$^{2+}$, which is even more

**Table 2.** Experimentally determined observed rate constants of the forward and backward chemical reactions for mutant forms of adenylate kinase (Adk) in the presence of Mg$^{2+}$.

Note that the corresponding rate constants for the phosphoryl transfer in the wild-type protein are too fast to be directly measured and have been estimated to be more than three orders of magnitude faster than in the mutants (*Kerns et al., 2015*).

| AAdk mutant | Rate constant (s$^{-1}$) 2 ADP → ATP + AMP | Rate constant (s$^{-1}$) ATP + AMP → 2 ADP |
|---|---|---|
| R124K | 4.6 ± 0.5 | 1.6 ± 0.4 |
| R150K | 1.3 ± 0.3 | 1.1 ± 0.3 |
| R161K | 0.4 ± 0.1 | 0.5 ± 0.2 |
| R85K | 0.3 ± 0.1 | 1.6 ± 0.2 |
| R36K | 0.8 ± 0.2 | 14 ± 2 |

pronounced with the optimal Mg²⁺ ion, is not new and has been well established in the literature for P-transfer reactions, including the Adk chemical reaction (*Kerns et al., 2015*; *Pérez-Gallegos et al., 2017*; *Rosta et al., 2014*; *Yang et al., 2012*). The striking new result, central to this work, is the large decrease in the entropic barrier in the presence of a divalent cation, thereby strongly supporting our key finding of a broader TSE from the QM/MM simulations.

We were able to further test our QM/MM simulations with a second designed set of experiments by measuring the pH dependence of the P-transfer step. We find that the reaction rate increases with higher pH in the presence of Ca²⁺, whereas without a metal the opposite trend is observed (*Figure 5B*). These experimental results match the simulation results: the fully charged nucleotides state was the most reactive in the presence of the metal, whereas the monoprotonated state had a much higher free-energy barrier. In contrast, without metal only the monoprotonated nucleotides were reactive. Notably, at low pH (pH 5), the P-transfer step becomes rate limiting even with Mg²⁺, allowing for an estimate of the rate enhancement of Mg²⁺ versus Ca²⁺ of 24-fold.

Finally, we measured the forward and backward reaction rate constants of several arginine mutants that were identified as important for the P-transfer from the QM/MM simulations: R36K, R85K, R124K, R150K, and R161K (*Table 2*). The large decrease in the catalytic rate for each single Arg to Lys mutation further highlights the notion of a highly choreographed active site for a well-coordinated P-transfer (*Figure 3G*), in which arginine residues that are not even directly coordinated to the transferring phosphate play an equally important role.

## Discussion

Adk is an impressively efficient enzyme, being able to accelerate the P-transfer reaction over a billion times. Previous work showed the complexity of the Adk free-energy landscape for this two-substrate, two-product reaction (*Figure 1A*), providing direct experimental evidence for the widely accepted concept of a multidimension free-energy landscape with multiple intermediates and transitions states for the full enzymatic cycle, already described and demonstrated by Benkovic and Hammes-Schiffer (*Benkovic et al., 2008*). Full lid-closure results in an excellent preorientation of the donor and acceptor groups of the two bound nucleotides and the aiding active-site residues in the enzyme/substrate complex, in agreement with the general view of necessary pre-organization in enzyme catalysis. Lid-opening had been identified as the rate-limiting step in the enzymatic cycle (*Kerns et al., 2015*), and the corresponding TSE for this conformational change (*Stiller et al., 2019*) and the high-resolution structure of next intermediate, the EP complex before product dissociation, recently been described (*Stiller et al., 2022*).

For a comprehensive understanding of the complete free-energy landscape of Adk catalysis, we here investigate the key step, catalyzing the actual chemical step of P-transfer. While ES and EP complexes such as Adk bound to ADP/Mg²⁺ can be structurally characterized by traditional experimental methods as they represent a minimum in the free-energy landscape, the reaction path from substrate to product involving breaking and forming covalent bonds (chemical step) and traversing the crucial TS can only be 'visualized' by quantum mechanics-based molecular simulations. The power of such simulations to examine the chemical steps in enzymes, including P-transfer reactions, has been extensively documented (*Jin et al., 2017*; *Lai and Cui, 2020b*; *Mokrushina et al., 2020*; *Pérez-Gallegos et al., 2015*; *Pérez-Gallegos et al., 2017*; *Roston and Cui, 2016*; *Valiev et al., 2007*). The focus in the literature has been on the comparison of the enzyme-catalyzed and uncatalyzed reaction with respect to the bond character at the transitions state (i.e., associate versus dissociative or tight versus loose) (*Admiraal and Herschlag, 1995*; *Hengge, 2002*; *Kamerlin et al., 2013*; *Lassila et al., 2011*). QM calculations for uncatalyzed reactions have been well documented in the literature showing narrow and symmetric TSE (*Klähn et al., 2006*: *Wang et al., 2015*).

Here, we uncover a central new result that provokes a modified TS theory. Enzymes, due to their macromolecular nature, provide a fundamentally different, advantageous way to catalyze these chemical reactions compared to the uncatalyzed reaction by employing a broad TSE. Many different molecular configurations can be accommodated in the TS region with comparable energies via collective motions, spanning a 1-Å range along the reaction coordinate for the transferring phosphate that travels a total distance of only 2 Å during the P-transfer. This features resemblance the now well-established conformational sampling of proteins in ground states such as ES complexes. Our findings explain why enzymes do not face an entropic bottleneck for catalysis. Furthermore, as

enzyme active sites are asymmetric in contrast to the symmetric nature of the solvent for uncatalyzed reactions, we find that the TSE comprises also highly asymmetric conformations. Our findings help to resolve the controversy about the nature of the TS in enzyme-catalyzed P-transfer reactions between theory and experiments (*Lassila et al., 2011*). The complex nature of the active site of enzymes, in contrast to simple solvent, results in different mechanisms in the enzyme-catalyzed reaction.

We note that a previous QM/MM minimum energy path calculations for a different Adk enzyme, *E. coli* Adk, using a semiempirical method had proposed a different mechanism with a stable metaphosphate intermediate, reporting an even lower energy for this metaphosphate intermediate than the ES and EP complexes (*Shibanuma et al., 2020*). In complex systems, such as enzymes, it is possible to observe artificial local minima when using minimum energy path searching strategies due to inadequate sampling (*Mendieta-Moreno, 2015*; *Quesne et al., 2016*). From our experimental NMR and X-ray data, we know that such a stable metaphosphate does not exist in Adk-catalyzed reactions, highlighting the importance of experimental verification of simulations as performed here, and the use of extensive sampling with proper thermodynamic treatment.

More recently, another group performed QM/MM simulations for *E. coli* Adk using the semiempirical AM1/d-phoT method (*Nam et al., 2007*) and umbrella sampling, and measured and computed rate reduction in the chemical step by similar Arg mutants as reported here (*Ojeda-May et al., 2021*; *Dulko-Smith et al., 2023*). In contrast to the paper described above (*Shibanuma et al., 2020*), their results are in agreement with our data. First, the experimentally measured rate reductions in *Ojeda-May et al., 2021*; *Dulko-Smith et al., 2023* are in full agreement with our results. Second, our activation barriers from umbrella sampling are qualitatively similar to their values, with the obvious differences in the enzyme species (we study the thermophilic Aquifex Adk), and their choice of analyzing the QM/MM simulations in the opposite direction ATP/AMP to 2 ADP molecules. The latter lead them to singly protonate one oxygen on ATP, since this 'back reaction' is faster for the monoprotonated state, in agreement with our simulations. This nice agreement in both experiments and computation by our two groups strengthens the validity of reported results.

Intriguingly, the Nam/Wolf-Watz team (*Ojeda-May et al., 2021*; *Dulko-Smith et al., 2023*) and our group focus on completely different interpretations of the data! Nam/Wolf-Watz propose an interconnection, synchronization, between the chemical step and lid-opening. This language can be misleading since it can imply that the chemical step directly effects the opening, which of course cannot be the case. There is no 'memory' of the chemical step, the energy from the closing/opening step is dissipated within picoseconds. Lid-opening/closing is an energetically fully independent step from the P-transfer step, see our *Figure 1A*. We agree with the obvious result/interpretation by the authors that the Arg residues in the active site are important for both the P-transfer and the consequent lid-opening, but there is no synchronization between these independent steps. In contrast, we discover here a fundamental concept for rate enhancement by an optimal enzyme, the reduction in the activation entropy by a wide TSE. New experiments were triggered by our finding that then delivered experimental validation of this concept.

A wide TSE agrees with the definition of the TS being a surface on the potential energy surface instead of a saddle as elegantly discussed in a theoretical paper by Nussinov and colleagues (*Ma et al., 2000*). Our results deliver concrete evidence for this fundamental concept from QM/MM calculations and commitment analysis on Adk that were further buttressed by the consequently designed experiments. Although we have only quantitatively demonstrated this TSE concept here for Adk-catalyzed P-transfer, we hypothesize such mechanism to be more general for enzyme catalysis as it is rooted in the macroscopic nature of proteins. A conformationally spread-out TSE is likely one reason why enzymes are much bigger than a few active-site residues: On the one hand residues remote from the site of the chemical action allow for conformational flexibility without the protein unfolding, on the other hand they allow for increasing the probability of highly choreographed motions along the reaction coordinate (*Hammes-Schiffer and Benkovic, 2006*; *Klinman and Kohen, 2013*; *Saen-Oon et al., 2008*; *Schramm and Schwartz, 2018*) while reducing excursions into unproductive conformations (*Otten et al., 2020*). The latter mechanism is directly demonstrated here in the TSE, as the configurations seen in the simulations are along the P-transfer reaction coordinate. We note that we focus here on equilibrium effects and an atomistic description of the TSE, this is not related to the fundamental work on dynamical effects manifested in transmission coefficients due to barrier recrossing (*Antoniou*

*and Schwartz, 2016*; *Zinovjev and Tuñón, 2017*). The latter has been quantitatively demonstrated to play a minor role.

A question often being asked is why would nature evolve a conformational step to be rate limiting, as the lid-opening step for Adk? We propose that this is the 'price' the enzyme pays for stabilizing the ES complex to enable efficient rate acceleration of the chemical step, and for suppressing detrimental alternate reactions such as phosphate hydrolysis. Release of product requires disassembly of such closed, low-energy structure. These principles have now been directly revealed on an atomic scale for Adk: extensive electrostatic interactions of five Arg and one Lys with the phosphoryl groups to efficiently accelerate the P-transfer step, which then need to be broken for lid-opening to allow product release. The effect of different Arg mutations on the lid-opening has been recently nicely demonstrated (*Ojeda-May et al., 2021*; *Dulko-Smith et al., 2023*).

As long as the rate of this conformational transition ($k_{open}$) and hence $k_{cat}$ is not limiting for organismal fitness, there is no evolutionary pressure to further enhance it.

The importance of TSE including multiple transition pathways for protein folding (*Frauenfelder et al., 2006*; *Royer, 2008*), and more recently for protein conformational transitions within the folded state (*Pontiggia et al., 2015*; *Stiller et al., 2019*; *Tsai and Nussinov, 2014*) has been well documented. TSE in enzyme catalysis of chemical steps where covalent bonds are broken and formed as demonstrated here further promotes a unifying concept of protein folding and function, as the same principal concept of TSE is seen for these different types of conformational changes. Consequently, protein folding and function is embedded in a unifying energy landscape which is simply altered by ligands and solvent.

## Materials and methods
### Computational setup of the system
The crystal structure of Adk from *A. aeolicus* in complex with Ap5A coordinated to $Zn^{2+}$ (PDB ID 2RGX; *Henzler-Wildman et al., 2007*) was used as model. The $Zn^{2+}$ ion was replaced by $Mg^{2+}$ and the ADPs structures were built by modifying the structure of Ap5A, keeping the crystallographic coordination of the cation (2 oxygens from ADPs and 4 waters). For the simulation in low pH and without the magnesium ion, the nucleophilic oxygen of AD(T)P was protonated. Standard protonation states were assigned to all titratable residues (aspartate and glutamate residues are negatively charged, lysines and arginines residues are positively charged). Histidine protonation was assigned favoring formation of hydrogen bonds in the crystal structure. Each protein was immersed in a truncated octahedral box of TIP3P water (*Jorgensen et al., 1983*) and neutralized with $Na^+$ ions (3–5 ions, depending on the system).

### Classical simulation parameters
The parameters used for all protein residues were those corresponding to AMBER force field ff99sb (*Hornak et al., 2006*). For the non-protein residues, such as ADP and $Mg^{2+}$, parameters were taken from *Meagher et al., 2003*; *Allnér et al., 2012*. All simulations were performed with periodic boundary conditions. Pressure and temperature were kept constant with the Berendsen barostat and thermostat, respectively (*Berendsen et al., 1984*). The SHAKE algorithm (*Ryckaert et al., 1977*) was used to keep hydrogen atoms at their equilibrium bond distances, and a 2-fs time step was employed, except for the system without $Mg^{2+}$ for which it was set to 1 fs. The ADP–ADP system was initially subjected to a classical equilibration protocol, followed by QM/MM equilibration and production simulations. Starting from the classically equilibrated ADP–ADP, the monoprotonated systems with ADPH–ADP was built, by adding the H+and removing a $Na^+$ from the solvent, and subsequently the system without $Mg^{2+}$ was built by removing the ion and adding two $Na^+$ to the solvent. Each of these systems was again equilibrated classically and then used for forward QM/MM simulations. The backward simulations were equilibrated just using QM/MM simulations because the starting point was the product of the forward SMD simulation with the lowest work at product RC. Classical equilibration followed consisted of: first, 2000 steps of conjugate gradient optimization, followed by 100 ps MD simulation at NVT condition, where the system's temperature was slowly raised to 300 K; the third step consisted of 100 ps MD simulation at NPT conditions to equilibrate the system's density. During the temperature and density equilibration process, the protein α-carbon atoms were restrained by a

1 kcal/mol harmonic potential. Finally, non-restrained classical simulation was performed at least for 1 ns at NPT conditions to obtain an equilibrated structure to be used as the QM/MM input.

**QM/MM simulations** were performed using a 1-fs time step without SHAKE (*Ryckaert et al., 1977*). To equilibrate the system with the QM/MM hamiltonian, we performed first a conjugate gradient QM/MM optimization, followed by a 50-ps QM/MM MD equilibration process at NVT condition, subsequently each production simulation was performed at NVT condition (at least 100 ps saving coordinates every 1 ps). QM/MM calculations were performed using the additive scheme implemented in the sander module of the AMBER program suite (*Case et al., 2005*). The quantum (QM) region was treated with the self-consistent charge-density functional tight-binding (SCC-DFTB) approach (*Elstner, 2007*), usually considered a semiempirical QM method, as implemented in Sander (*de M Seabra et al., 2007*; *Walker et al., 2008*). The SCC-DFTB parameter set involving carbon, hydrogen, oxygen, nitrogen, phosphate, and magnesium was employed throughout this study (*Cai et al., 2007*; *Yang et al., 2008*). All the other atoms were described with AMBER ff99SB force field (*Hornak et al., 2006*). The interface between the QM and MM portions of the system was treated with the scaled position-link atom method. The electrostatic interactions between the QM and MM regions were treated using electrostatic embedding using a simple mulliken charge–restrained electrostatic potential (RESP) charge interaction. In the case of the van der Waals interactions, these were calculated classically.

The QM region of the complete system consists of the diphosphate moiety of both ADP molecules, one $Mg^{2+}$ ion, and the four water molecules that coordinate the metal ion in the crystal structure (*Figure 1—figure supplement 1*).

## Free-energy determination strategy using MSMD and Jarzynski's equation

The FEPs were constructed by performing constant velocity MSMD simulations (i.e., stiff string approximation), and using Jarzynski's equality (*Jarzynski, 1997*), which relates equilibrium free-energy values with the irreversible work performed over the system along a user defined reaction coordinate that drives the system from reactants to products. In the present study, the reaction coordinate ($\xi$) was chosen as:

$$\xi = d(O_{leaving} - P) - d(O_{attacking} - P) \tag{1}$$

Calculations were performed using a force constant of 300 kcal/mol/Å and pulling velocities of 0.05 Å $ps^{-1}$. To reconstruct the FEP of the phosphate transfer reaction, two sets of at least 10 SMD runs were performed starting from equilibrated QM/MM MD structures corresponding to either the system in (1) the Adk/ADP–ADP state (presently defined as forward reaction), and (2) the Adk/ATP–AMP state (backward reaction). Each set of work profiles was used to obtain the corresponding forward and backward FEPs with Jarzynski's equality (*Jarzynski, 1997*), and finally the two profiles were combined to obtain the complete curve. Since when using MSMD in combination with Jarzynski's inequality, the system starts from equilibrium conditions and is driven along the RC further from equilibrium as the reaction proceeds, and therefore the FEP estimate increasingly overestimates the real FEP. The best way to combine forward and backward reactions is performed by keeping the initial segment (and thus lower) FEP of each forward and backward reaction estimates. This strategy was successfully used to obtain enzymatic reaction FEPs in previous works (*Crespo et al., 2005*; *Defelipe et al., 2015*).

Along the present work, the ADP in the ATP-lid will be called as AD(T)P and the other, in the AMP-lid, as AD(M)P. The reaction involves the two P–O bonds, one forming and the another breaking, the two bonds sharing the same phosphorus atom from the transferring phosphate (*Figure 1B*). In the former, an oxygen atom of the AD(T)P reacts and we will name it as $O_{attacking}$. The breaking bond is described by the oxygen atom for the AD(M)P and it will be called $O_{leaving}$.

## Higher-level DFT(PBE) free-energy calculations

The FEP of the ADK with $Mg^{2+}$ was also computed using a higher level of theory. Here, the QM system was described with PBE functional (*Perdew et al., 1996*) using a DZVP basis set as implemented in the GPU-based code LIO that works with amber (*Nitsche et al., 2014*). To take advantage of previous DFTB sampling, five lowest work versus RC profiles of the forward and backward reactions were

selected, and segmented in 10 windows or stages. For each window, the FEP at the DFT(PBE) level was computed using SMD with a steering velocity of 0.05 Å/ps. Windows were combined to obtain both forward and backward FEP and the final profile was obtained by joining them. Details of the staged strategy can be found in references (*Ozer et al., 2010*; *Ozer et al., 2012*; *Ganguly et al., 2020*).

## Commitment analysis

To verify the true nature of the observed TS we performed a commitment analysis. We first selected 20 structures for each of 11 points along the RC in the TS region (every 0.1 Å from −0.6 to 0.6 Å on the RC). Structures were equilibrated at the corresponding RC position for 5 ps using a harmonic restrain potential of 300 kcal/mol/Å. From the final equilibrated structure, we used the modified Berendsen thermostat (*Bussi et al., 2007*) to assign random initial velocities, and performed 10 non-restrained simulations. Each simulation was followed and determined whether it reached the reactants or products. Finally, the resulting probability (estimated from the observed frequency) to reach product for each point along the RC was computed.

## Umbrella sampling simulations

Finally, to avoid possible bias due to the use of MSMD to obtain the FEP, we also computed the corresponding profile using umbrella sampling. The initial structures for each window were randomly selected from the lowest work MSMD trajectories along the RC. The force constant for the harmonic restraint was set to 300 kcal/mol/Å$^2$, ensuring sufficient confinement without inducing artificial behavior. Windows were separated every 0.1 Å along the whole RC. Each window simulation started with a equilibration run of 60 ps at NPT condition, followed by sampling over a production run of 40 ps, with coordinates recorded at 1 ps intervals and the reaction coordinate values every 1 fs. Simulations were performed at the previously described SCC-DFTB level of theory. The weighted histogram analysis method was employed to reconstruct the free-energy landscape from the individual window simulations (*Kumar et al., 1992*; *Grossfield, 2010*).

## Principal component analysis

One single trajectory containing the TSE structures of MSMD and umbrella sampling was created by aligning the atoms of the P–O$_{leaving}$ and P–O$_{attacking}$ to a reference. The reference was the reactant structure (ADP/ADP). This was done in order to compare the TSEs using the same coordinate system. The principal component analysis was calculated over the transferring phosphate atoms, Mg$^{2+}$ ion, P$_\alpha$ and O3α atoms from AD(M)P and P$_\beta$ and O3β atoms from AD(T)P. Those atoms were selected because they are directly involved in the chemical reaction.

## Steady-state kinetics measurements

Steady-state kinetics measurements for *A. aeolicus* Adk (Adk) were performed at different pH values in the absence of divalent metals (using 50 mM of EDTA) and the presence of Mg$^{2+}$ or Ca$^{2+}$. In all cases, the reaction was started by the addition of 8 mM ADP and saturating metal concentrations were used if applicable (32 mM at pH 5 and 8 mM at pH 7 and 9, respectively). The enzyme concentration varied between 30 nM and 200 μM depending on the rate of interconversion; measurements were collected at 25°C. The amount of product was quantified with high-pressure liquid chromatography (HPLC). Protein precipitated by quench (30% Trichloroacetic acid,TCA, + 6 M HCl mixture) was separated with Spin-X centrifugal tube filters (Costar), filtered supernatant was diluted to avoid HPLC detector saturation, and the pH was brought to 6.0 to achieve optimal separation. The samples were analyzed on an HPLC system (Agilent Infinity 1260) with a high-precision autosampler (injection error <0.1 μl) and analytical HPLC column ACE (i.d. 2.4 mm, length 250 mm, C18-AR, 5 Å pore size) and separated with isocratic elution with potassium phosphate mobile phase (100 mM, pH 6.1). The observed rate constants were determined from 8 to 15 data points for each temperature and/or pH value using initial rate analysis. The values and uncertainties (SD) shown in *Figure 5* and *Table 2* were determined from least-squares linear regression. Similarly, values and uncertainties (SD) of Δ$H^\ddagger$ and Δ$S^\ddagger$ were extracted from linear regression of the data points presented in *Figure 5A*.

## Temperature dependence of catalysis and mutant activity

Experiments were essentially performed as described previously (*Kerns et al., 2015*) and above. In short, the steady-state kinetics measurements were collected at temperatures between 20 and

60°C for the temperature dependency and at 25°C for the mutants. For the temperature dependence, the samples contained 4 mM ADP and equimolar (with nucleotide) concentrations of calcium or 50 mM EDTA. The enzyme concentration was varied between 50 nM to 1 μM (with $Ca^{2+}$) and 100 to 200 μM (with EDTA); buffer was 100 mM HEPES, pH 7.0, and 80 mM KCl. The amount of product produced over time (10–16 min) was quantified with HPLC and observed rates were extracted as described above. A similar approach was used for the Arg-to-Lys Aadk mutants, where the reaction was measured in both directions starting with either 4 mM ADP or ATP/AMP and equimolar (with nucleotide) concentrations of $Mg^{2+}$.

## Acknowledgements

This work was supported by the Howard Hughes Medical Institute (HHMI) to DK. We thank to 'High-Performance Computing Center' (CeCAR, https://cecar.fcen.uba.ar/) of the *Facultad de Ciencias Exactas y Naturales* at the University of Buenos Aires for the computational resources. We are grateful to the Brazilian Biosciences National Laboratory (LNBio), part of the Brazilian Center for Research in Energy and Materials (CNPEM) for accessibility to the High-Performance Computing Cluster and scientific infrastructure.

## Additional information

### Competing interests

Dorothee Kern: D.K. is co-founder of Relay Therapeutics and MOMA Therapeutics. The other authors declare that no competing interests exist.

### Funding

| Funder | Grant reference number | Author |
| --- | --- | --- |
| Howard Hughes Medical Institute | | Dorothee Kern |

The funders had no role in study design, data collection and interpretation, or the decision to submit the work for publication.

### Author contributions

Gabriel E Jara, Formal analysis, Investigation, Visualization, Methodology, Writing – original draft, Writing – review and editing; Francesco Pontiggia, Renee Otten, Roman V Agafonov, Data curation, Formal analysis, Investigation, Visualization; Marcelo A Martí, Conceptualization, Resources, Supervision, Investigation, Writing – original draft, Project administration, Writing – review and editing; Dorothee Kern, Conceptualization, Resources, Supervision, Funding acquisition, Investigation, Methodology, Writing – original draft, Project administration, Writing – review and editing

### Author ORCIDs

Gabriel E Jara ⬥ https://orcid.org/0000-0002-5831-1392
Marcelo A Martí ⬥ http://orcid.org/0000-0002-7911-9340
Dorothee Kern ⬥ https://orcid.org/0000-0002-7631-8328

Reviewer #1 (Public Review): https://doi.org/10.7554/eLife.93099.4.sa1
Reviewer #2 (Public Review): https://doi.org/10.7554/eLife.93099.4.sa2
Author response https://doi.org/10.7554/eLife.93099.4.sa3

## Additional files

### Supplementary files

MDAR checklist

## Data availability

All the computational data produced in the current study is available in the following repository: ZENODO: https://doi.org/10.5281/zenodo.14647770.

The following dataset was generated:

| Author(s) | Year | Dataset title | Dataset URL | Database and Identifier |
|---|---|---|---|---|
| Jara G, Marti MA, Kern D, Pontiggia F, Otten R, Agafonov R | 2025 | Wide Transition-State Ensemble as Key Component for Enzyme Catalysis | https://doi.org/10.5281/zenodo.14647770 | Zenodo, 10.5281/zenodo.14647770 |

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
