## [Editor Report · eLife Assessment]

In this potentially **important** study, the authors report results of QM/MM simulations and kinetic measurements for the phosphoryl-transfer step in adenylate kinase. The results point to the mechanistic proposal that the transition state ensemble is broader in the most efficient form of the enzyme (i.e., in the presence of Mg2+ in the active site) and thus a different activation entropy. With a broad set of computations and experimental analyses, the level of evidence is considered **solid** by some reviewers. On the other hand, there remain limitations in the computational analyses, especially regarding free energy profiles using different methodologies (shape of free energy profiles with DFTB vs. PBE QM/MM, and barriers with steered MD and umbrella sampling) and the activation entropy, leading some reviewers to the evaluation that the level of evidence is **incomplete**.

---

## [Referee Report · Reviewer #1 (Public Review)]

Summary:

This study investigated the phosphoryl transfer mechanism of the enzyme adenylate kinase, using SCC-DFTB quantum mechanical/molecular mechanical (QM/MM) simulations, along with kinetic studies exploring the temperature and pH dependence of the enzyme's activity, as well as the effects of various active site mutants. Based on a broad free energy landscape near the transition state, the authors proposed the existence of wide transition states (TS), characterized by the transferring phosphoryl group adopting a meta-phosphate-like geometry with asymmetric bond distances to the nucleophilic and leaving oxygens. In support of this finding, kinetic experiments were conducted with Ca2+ ions at different temperatures and pH, which revealed a reduced entropy of activation and unique pH-dependence of the catalyzed reaction.

Strengths:

A combined application of simulation and experiments is a strength.

Weaknesses:

The conclusion that the enzyme-catalyzed reaction involves a wide transition state is not sufficiently clarified with some concerns about the determined free energy profiles compared to the experimental estimate. (See Recommendations for the authors.)

Comments on revisions:

While the authors have made some improvements in clarifying the manuscript, questions still remain about their conclusion regarding the wide-TS, which appears this may be a misinterpretation of the simulation results. Also, they should clearly point out the large discrepancies between DFTB QM/MM and PBE QM/MM results (shape of free energy files) and also between steered MD and umbrella sampling results (barriers). Another question is the large change in activation entropy (between the reaction with and without divalent cations). This difference may be difficult to attribute sorely to the difference in the reaction geometries near TS.

---

## [Referee Report · Reviewer #2 (Public Review)]

Summary:

The authors report results of QM/MM simulations and kinetic measurements for the phosphoryl-transfer step in adenylate kinase. The main assertion of the paper is that a wide transition state ensemble is a key concept in enzyme catalysis as a strategy to circumvent entropic barriers. This assertion is based on observation of a "structurally wide" set of energetically equivalent configurations that lie along the reaction coordinate in QM/MM simulations, together with kinetic measurements that suggest a decrease of the entropy of activation.

Strengths:

The study combines theoretical calculations and supporting experiments.

Weaknesses:

The current paper hypothesizes a "wide" transition state ensemble as a catalytic strategy and key concept in enzyme catalysis. Overall, it is not clear the degree to which this hypothesis is fully supported by the data. The reasons are as follows:

(1) Enzyme catalysis reflects a rate enhancement with respect to a baseline reaction in solution. In order to assert that something is part of a catalytic strategy of an enzyme, it would be necessary to demonstrate from simulations that the activation entropy for the baseline reaction is indeed greater and the transition state ensemble less "wide". Alternatively stated, when indicating there is a "wide transition state ensemble" for the enzyme system - one needs to indicate that is with respect to the non-enzymatic reaction. However, these simulations were not performed and the comparisons not demonstrated. The authors state "This chemical step would take about 7000 years without the enzyme" making it impossible to measure; nonetheless, the simulations of the nonenzymatic reaction would be fairly straightforward to perform in order to demonstrate this key concept that is central to the paper. Rather, the authors examine the reaction in the absence of a catalytically important Mg ion.

(2) The observation of a "wide conformational ensemble" is not a quantitative measure of entropy. In order to make a meaningful computational prediction of the entropic contribution to the activation free energy, one would need to perform free energy simulations over a range of temperatures (for the enzymatic and non-enzymatic systems). Such simulations were not performed, and the entropy of activation was thus not quantified by the computational predictions. The authors instead use a wider TS ensemble as a proxy for larger entropy, and miss an opportunity to compare directly to the experimental measurements.

Comments on revisions:

Overall, I do not think the authors have been able to quantitatively support their conclusion, and the qualitative support is somewhat weak. This makes the interpretation of the computational results somewhat speculative. Nonetheless, comparison was made for models with and without divalent ions, and the experimental data is valuable.

---

## [Author Response]

The following is the authors’ response to the previous reviews.

**Public Reviews:**

**Reviewer #1 (Public Review):**
Summary:This study investigated the phosphoryl transfer mechanism of the enzyme adenylate kinase, using SCC-DFTB quantum mechanical/molecular mechanical (QM/MM) simulations, along with kinetic studies exploring the temperature and pH dependence of the enzyme's activity, as well as the effects of various active site mutants. Based on a broad free energy landscape near the transition state, the authors proposed the existence of wide transition states (TS), characterized by the transferring phosphoryl group adopting a meta-phosphate-like geometry with asymmetric bond distances to the nucleophilic and leaving oxygens. In support of this finding, kinetic experiments were conducted with Ca2+ ions at different temperatures and pH, which revealed a reduced entropy of activation and unique pH-dependence of the catalyzed reaction.Strengths:A combined application of simulation and experiments is a strength.Weaknesses:The conclusion that the enzyme-catalyzed reaction involves a wide transition state is not sufficiently clarified with some concerns about the determined free energy profiles compared to the experimental estimate. (See Recommendations for the authors.)
**Reviewer #2 (Public Review):**
Summary:The authors report results of QM/MM simulations and kinetic measurements for the phosphoryl-transfer step in adenylate kinase. The main assertion of the paper is that a wide transition state ensemble is a key concept in enzyme catalysis as a strategy to circumvent entropic barriers. This assertion is based on observation of a "structurally wide" set of energetically equivalent configurations that lie along the reaction coordinate in QM/MM simulations, together with kinetic measurements that suggest a decrease of the entropy of activation.

Thank you for your feedback. The reviewer’s questions are answered below, hoping to clarify them.

Strengths:The study combines theoretical calculations and supporting experiments.Weaknesses:The current paper hypothesizes a "wide" transition state ensemble as a catalytic strategy and key concept in enzyme catalysis. Overall, it is not clear the degree to which this hypothesis is fully supported by the data. The reasons are as follows:(1) Enzyme catalysis reflects a rate enhancement with respect to a baseline reaction in solution. In order to assert that something is part of a catalytic strategy of an enzyme, it would be necessary to demonstrate from simulations that the activation entropy for the baseline reaction is indeed greater and the transition state ensemble less "wide". Alternatively stated, when indicating there is a "wide transition state ensemble" for the enzyme system - one needs to indicate that is with respect to the non-enzymatic reaction. However, these simulations were not performed and the comparisons not demonstrated. The authors state "This chemical step would take about 7000 years without the enzyme" making it impossible to measure; nonetheless, the simulations of the nonenzymatic reaction would be fairly straight forward to perform in order to demonstrate this key concept that is central to the paper. Rather, the authors examine the reaction in the absence of a catalytically important Mg ion.

Thank you for your thoughtful feedback. QM/MM calculations for uncatalysed phosphoryl-transfer reactions involving either diphosphates or triphosphates have been well documented in the literature showing narrow and symmetric TSE (Klan et al., JACS 2006, 128 (47) 15310-15323; Cui Wang et al., JPCB 2015, 119(9), 3720-3726). We added these references to the revised manuscript.

(2) The observation of a "wide conformational ensemble" is not a quantitative measure of entropy. In order to make a meaningful computational prediction of the entropic contribution to the activation free energy, one would need to perform free energy simulations over a range of temperatures (for the enzymatic and non-enzymatic systems). Such simulations were not performed, and the entropy of activation was thus not quantified by the computational predictions. The authors instead use a wider TS ensemble as a proxy for larger entropy, and miss an opportunity to compare directly to the experimental measurements.

Although we share the reviewers desire to quantify entropies from QM/MM simulations, we agree with discussions in the literature that calculating quantitative entropies from performing QM/MM simulations at different temperatures is not reliable. We therefore felt strongly to stay with a qualitative assessment of entropy differences from our simulations. As the reviewer highlighted, our study combines theoretical calculations and experiments. The entropy of activation is well estimated by the experiments from the experimental accuracy of these temperature-dependent changes in rate constants for the chemical step. Our computational results agree well with the experimental results and were further validated in 2 rounds of reviews/revisions by additional different free energy calculation methods (MSMD and US), plus committor analysis.

**Reviewer #3 (Public Review):**
Summary:By conducting QM/MM free energy simulations, the authors aimed to characterize the mechanism and transition state for the phosphoryl transfer in adenylate kinase. The qualitative reliability of the QM/MM results has been supported by several interesting experimental kinetic studies. However, the interpretation of the QM/MM results is not well supported by the current calculations.

Thank you for your feedback. We appreciate the recognition of our experimental validation but understand your concern about the interpretation of our QM/MM results. To address this, we answer the specific questions below and added clearer explanations of the computational approach, including its limitations. We also better aligned the QM/MM results with both experimental data and theoretical expectations to strengthen the overall interpretation.

Strengths:The QM/MM free energy simulations have been carefully conducted. The accuracy of the semi-empirical QM/MM results was further supported by DFT/MM calculations, as well as qualitatively by several experimental studies.Weaknesses:(1) One key issue is the definition of the transition state ensemble. The authors appear to define this by simply considering structures that lie within a given free energy range from the barrier. However, this is not the rigorous definition of transition state ensemble, which should be defined in terms of committor distribution. This is not simply an issue of semantics, since only a rigorous definition allows a fair comparison between different cases - such as the transition state in an enzyme vs in solution, or with and without the metal ion. For a chemical reaction in a complex environment, it is also possible that many other variables (in addition to the breaking and forming P-O bonds) should be considered when one measures the diversity in the conformational ensemble.In the revised manuscript, the authors included committor analysis. However, the discussion of the result is very brief. In particular, if we use the common definition of the transition state ensemble (TSE) as those featuring the committor around 0.5, the reaction coordinate of the TSE would span a much narrower range than those listed in Table 1. This point should be carefully addressed.

The reviewer is right, the TSE is rigorously defined in terms of the committor distribution. We actually calculated the committor distribution for the reaction with and without Mg. We now added the figure showing the committor distribution for both reactions (Figure 3 – supplement 9). We did not include these results before because the committor distribution histogram would need more points to have a more accurate shape, requiring a high computational cost. We followed the reviewer’s suggestion and updated table 1 with the values from the committor distribution analysis.

(2) While the experimental observation that the activation entropy differs significantly with and without the Ca2+ ion is interesting, it is difficult to connect this result with the "wide" transition state ensemble observed in the QM/MM simulations so far. Even without considering the definition of the transition state ensemble mentioned above, it is unlikely that a broader range of P-O distances would explain the substantial difference in the activation entropy measured in the experiment. Since the difference is sufficiently large, it should be possible to compute the value by repeating the free energy simulations at different temperatures, which would lead to a much more direct evaluation of the QM/MM model/result and the interpretation.

See our answer above about this point.

**Recommendations for the authors:**

**Reviewer #1 (Recommendations For The Authors):**
Major comments:One of the remaining issues with this revision is the assertion of the wide transition states in the presence of Mg2+ ion. When discussing the transition state of phosphoryl transfer reactions, it is important to consider their nature as involving both the cleavage and formation of P-O bonds. While these two events can occur in concert with a single transition state, many studies have shown that one event often precedes the other. Sometimes, there is a slight drop in free energy between the two events, forming a transient intermediate. However, due to its very short lifetime, this intermediate state may not be detectable experimentally. Depending on the sequence of events, the transition state or the transient intermediate may exhibit characteristics of a metaphosphate or phosphorane-like species. Based on the DFTB simulation results presented in the paper, it appears that the reaction forms a metaphosphate-like transition state. In the present reaction, since the two oxygen atoms involved in the reaction are very good leaving groups with similar reactivity, it is not surprising to observe the two events near the TS with very similar relative free energies, and therefore, the free energy profile can be very flat near the TS. This is consistent with the statement that "the transferring phosphate can be much closer to the leaving oxygen than the attacking oxygen and vice versa" on page 9. In my opinion, however, this should not be considered as a wide transition state but rather a consequence of the two events occurring very close to each other along the reaction coordinate. This distinction can be considered a semantic issue, and as long as the authors clearly discuss this issue and clarify the meaning of the TS ensemble, the reviewer is okay with that. In its current form, the statement of the wide TS ensemble may lead to a misleading interpretation of the reaction under study.An intermediate is clearly defined as a minimum in the free energy landscape. We see no evidence in any of your simulations of a minimum flanked by two transitions states, nor do we see any evidence in our NMR relaxation data or crystal structure ensemble refinement. We report our experimental and computational results, so that the reader can directly interpret the free energy landscapes for this system, avoiding semantics due to language ambiguity.Second, based on the kinetic study, the free energy of the catalytic reaction is approximately zero. The authors suggest that at pH near 7, the ADP exists as a roughly50-50 mixture between the singly protonated and fully charged states and consequently, the reaction free energies between the two scenarios cancel each other out. However, this argument is not correct. If [ADP(H)]/[ADP] is close to 1, the two reaction free energies, one with +6 kcal/mol and the other with -6 kcal/mol, imply that the protonation of the products (either ATP or AMP) requires ~12 kcal/mol (i.e., 9 pKa unit shift). Given the symmetric nature of the reaction and the similar pKa values between ATP, ADP versus AMP, such a large shift in the pKa of the product state is not expected, and for the calculated results to be accurate, the pKa shifts in the reactant state versus the product state must be opposite, with a total relative shift of 9 pKa units. This is difficult to understand given the nature of the reaction catalyzed by the adenylate kinase enzyme.

We thank this reviewer for this question, which made us realize that we cannot compare the free energies of our QM/MD simulations with the experimentally determined ADP and ATP/AMP ratios. In the experiment we determine the entire pool of ADP and AMP/ATP bound to the enzyme, but could not distinguish if the protonated and or nonprotonated states are contributing to the measured observed rate constants Kerns, S. et al.,(2015). In the present study, we now discovered that the nonprotonated forms have a lower activation barrier, but the protonated states also contribute to the overall reaction. Therefore, we removed this paragraph from our discussion.

Minor comments:The difference in the free energy barrier determined by the MSMD and umbrella sampling is not negligible. Considering that umbrella sampling is commonly used in this type of research, the MSMD method appears to overestimate the barrier by more than 3 kcal/mol. Would the TS geometries obtained by umbrella sampling be comparable to those obtained by MSMD?

This is an excellent suggestion, since the umbrella sampling is the more accurate method. The TSE from both methods are indeed comparable, and we added new figure panels about this results to Fig. 4.

Figure 5 shows that the enthalpy of activation is similar for reactions with and without Ca2+. Do the authors expect the enthalpy of activation to decrease when Ca2+ is replaced by Mg2+ without a significant change in the entropy of activation? Any justification?

In Kerns, S. et al.,(2015) we had experimentally determined the dependence of the observed rate of the P-transfer on the nature of the divalent metal, with Mg2+ being by far superior to the other divalent metals. We proposed that this majorly is an effect on the enthalpy of activation, that other divalent metals provide poor orbital overlap, in agreement with published work on P-transfer reactions that show selectivity for a specific metal.

Please provide proper citations for SHAKE and WHAM.

The citations were added.

**Reviewer #2 (Recommendations For The Authors):**
The authors did not really address in the revised manuscript the main points of the previous review, which included examination of non-enzymatic reaction (via simulation, not measurement) and quantification of the connection between the reported wide TS ensemble and the increase in entropy (by additional simulations). The authors should also add reference to the AM1/d-PhoT model of Nam et al. which is now discussed.

QM/MM calculations for uncatazlysed phosphoryl-transfer reactions involving either diphosphates or triphosphates have been well documented in the literature showing narrow and symmetric TSE (Klahn et al., JACS 2006, 128 (47) 15310-15323; Cui Wang et al., JPCB 2015, 119(9), 3720-3726). We added these references to the revised manuscript.

The reference to AM1/d-PhoT model was added.

**Reviewer #3 (Recommendations For The Authors):**
In the revised ms, the authors indeed addressed many of the points raised in the previous round of review. In addition to the issue of TSE and committor mentioned above, another point that needs to be carefully explained is the very significant difference between umbrella sampling results and those in Fig. 1C - especially for the case without Mg2+ - the difference of more than 20 kcal/mol is not something that can be ignored at a qualitative level.

We thank the reviewer for pointing out that the difference in free energy profiles between umbrella sampling (US) and MSMD, especially in the case without Mg^2^+ needs to be addressed.

We believe that the key reason for this difference lies in the methodological approaches of these techniques.

Umbrella sampling is an equilibrium enhanced sampling method, that allows for a balanced and thorough exploration of the free energy landscape, the MSMD is a non-equilibrium method and estimation depends of the averaging scheme used and the number of trajectories. In the present work, the free energy was estimated using an exponential average. This averaging scheme has a slow convergence, small variance and may overestimate the free energy barrier, specially if the barrier as seen in the absence of Mg is quite high. This factor could explain the significant difference between umbrella sampling and MSMD combined with Jarzynski’s equality.

We have added new panels to Fig. 4 to compare the TSE from the more accurate umbrella sampling to the MSMD simulations, buttressing the validity of our original findings. We revised the manuscript discuss the differences between the MSMD and the umbrella sampling free energy profiles.